# A Multi-Objective Trajectory Planning Method for Collaborative Robot

**Jiangyu Lan** [1] , **Yinggang Xie** [1,*], **Guangjun Liu** [2] **and Manxin Cao** [1]

[1]   School of Information and Communication Engineering, Beijing Information Science and Technology
     University, Beijing 100101, China; lanjiangyu@mail.bistu.edu.cn (J.L.); cao_mx@mail.bistu.edu.cn (M.C.)
[2]   Department of Aerospace Engineering, Ryerson University, Toronto, ON M5B 2K3, Canada; gjliu@ryerson.ca
*   Correspondence: xieyinggang@bistu.edu.cn

**Abstract:** Aiming at the characteristics of high efficiency and smoothness in the motion process of collaborative robot, a multi-objective trajectory planning method is proposed. Firstly, the kinematics model of the collaborative robot is established, and the trajectory in the workspace is converted into joint space trajectory using inverse kinematics method. Secondly, seven-order B-spline functions are used to construct joint trajectory sequences to ensure the continuous position, velocity, acceleration and jerk of each joint. Then, the trajectory competitive multi-objective particle swarm optimization (TCMOPSO) algorithm is proposed to search the Pareto optimal solutions set of the robot's time-energy-jerk optimal trajectory. Further, the normalized weight function is proposed to select the appropriate solution. Finally, the algorithm simulation experiment is completed in MATLAB, and the robot control experiment is completed using the Robot Operating System (ROS). The experimental results show that the method can achieve effective multi-objective optimization, the appropriate optimal trajectory can be obtained according to the actual requirements, and the collaborative robot is actually operating well.

**Keywords:** collaborative robot; trajectory planning; B-spline; multi-objective optimization

## 1. Introduction

Industrial robots have been used widely in industrial production fields. However, with the development of society, traditional industrial robots have been unable to meet people's needs for safe collaboration and flexible deployment. Robots are required to handle more complex tasks and have more flexible and precise performance [1]. Collaborative robot (cobot) is one of the most popular research directions in the field of robotics [2]. Compared with traditional industrial robots working in fences, collaborative robots can work together with humans and complete various complex tasks in more scenes [3]. Therefore, collaborative robots need to meet two requirements: (1) higher operate efficiency and smooth motion to adapt to the requirements of various complex tasks; (2) a simpler and faster trajectory planning method to facilitate human-robot collaboration.

Trajectory planning refers to planning the trajectory of end-effector and joints of the robot in order to generate the reference input of the control system [4]. Firstly, the robot uses the interpolation algorithm to establish the time-trajectory sequence of each axis, including the trajectory position, speed, acceleration and jerk information. Then, under the constraints of kinematic performance of the robot, trajectory optimization is performed for efficiency, energy consumption, smoothness and other factors [5]. Different optimization schemes are selected according to the actual task situation.

In the research of trajectory interpolation, early researchers proposed a quintic polynomial interpolation method [6] for trajectory interpolation. This method has a small amount of calculation, but it is prone to distortion. With the enhancement of hardware performance, researchers began to

use spline curve to do robot trajectory planning. Alessandro et al. [7] proposed a trajectory planning method of cubic spline curve. This algorithm has fewer constraints and a faster calculation speed, but the acceleration curve has jitter, resulting in greater wear of the robot. Kong et al. [8] proposed cubic b-spline trajectory interpolation, the trajectory curve is relatively smooth, but the initial and final acceleration and jerk cannot be specified by themselves. Kong et al. [8] proposed a cubic b-spline trajectory interpolation method, which obtained a relatively smooth trajectory curve, but could not independently specify the initial and final values of acceleration and jerk.

In the research of trajectory optimization, researchers mainly focus on the optimization of time, energy consumption, and smoothness [9]. Xidias et al. [10] used a genetic algorithm with multiple populations to perform time-optimal trajectory planning for super-redundant manipulators. Luo et al. [11] used Lagrange interpolation and iterative methods for energy optimal trajectory planning. Lin et al. [12] used particle swarm optimization and clustering algorithms for jerk optimal trajectory planning. However, the above single-objective trajectory optimization is difficult to meet the application requirements of complex situations, many researchers begin to pay attention to multi-objective trajectory optimization. The traditional method transforms the multi-objective problem into the single-objective problem, and then uses the single-objective optimization method. Gasparetto et al. [13] transformed the time-jerk objective into a single objective using weight coefficients, and used sequence quadratic programming (SQP) to achieve trajectory optimization. However, this method is difficult to distribute weights reasonably, the diversity of solutions is insufficient, and it may fall into a local optimal solution. For the above problems, multi-objective evolutionary algorithm is an effective method. The multi-objective optimization algorithm can simultaneously optimize multiple objectives to obtain a set of Pareto optimal solutions, and then select the appropriate solution according to the specific situation. In Reference [14], the NSGA-II algorithm was used to obtain a set of Pareto optimal solutions, and an appropriate solution was selected for trajectory optimization.

Based on the above research, this paper proposed a simple and effective multi-objective trajectory planning method. In this paper, the higher-order B-spline interpolation method is chosen to replace the traditional interpolation method, so that the planned trajectory is smoother. Then, we selected three crucial objectives in the operation of the robot, and used the improved multi-objective optimization algorithm proposed in this paper to obtain the Pareto optimal solution set. For different scenarios (such as human-robot collaboration), this paper proposed the normalized weight function that can set different weight coefficients and select the most appropriate solution in the optimal solution set. Finally, in order to verify the effect of the method, we designed a trajectory editing software and imported the optimized trajectory into ROS system for actual control of the robot. The experimental results show that the obtained trajectory can make the robot run well. And the method can be easily used in various types of robot manipulators.

The rest of the paper is organized as follows. Section 2 conducts kinematic analysis of collaborative robots. Section 3 constructs a time-energy-jerk optimization mathematical model, uses seven-order B-spline functions for joint trajectory interpolation, and designs trajectory competitive multi-objective particle swarm optimization (TCMOPSO) for trajectory optimization. Section 4 completes the experiment of multi-objective trajectory planning. Section 5 is the summary and prospect of the work.

## 2. Kinematics Analysis of Collaborative Robot

### 2.1. Collaborative Robot Modeling

This section models the collaborative robot AUBO-I5 in the laboratory and performs forward and inverse kinematic analysis. AUBO-I5 [15] is a lightweight collaborative robot with six rotating joints, including base, shoulder, elbow and three wrist joints. The structure and dimensions of the robot are shown in Figure 1.

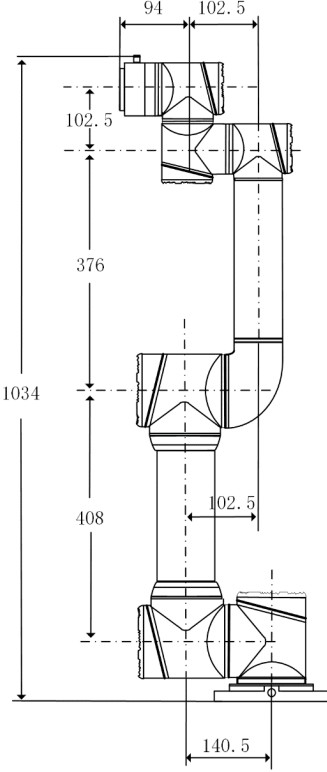

**Figure 1.** AUBO-I5 structure diagram.

The modified Denavit–Hartenberg (DH) method [16] was used to establish the kinematics model of the robot, as shown in Table 1.

**Table 1.** Modified DH parameters.

| Link i | $a_{i-1}$ (mm) | $\alpha_{i-1}$ (°) | $d_i$ (mm) | $\theta_i$ (°) | Range (°) |
|---|---|---|---|---|---|
| 1 | 0 | 0 | 98.5 | $\theta_1$ | ±175 |
| 2 | 0 | −90 | 121.5 | $\theta_2$ | ±175 |
| 3 | 408 | 180 | 0 | $\theta_3$ | ±175 |
| 4 | 376 | 180 | 0 | $\theta_4$ | ±175 |
| 5 | 0 | −90 | 102.5 | $\theta_5$ | ±175 |
| 6 | 0 | 90 | 94 | $\theta_6$ | ±175 |

In addition, in order to ensure that the performance index of the robot is met during operation, the kinematic constraints of each joint of the robot are given, as shown in Table 2.

**Table 2.** Kinematic constraints.

| Link i | $v_{max}/((°)\cdot s^{-1})$ | $a_{max}/((°)\cdot s^{-2})$ | $j_{max}/((°)\cdot s^{-3})$ | $\tau_{max}/(N\cdot m)$ |
|---|---|---|---|---|
| 1 | 148 | 1480 | 5920 | 376 |
| 2 | 148 | 1480 | 5920 | 376 |
| 3 | 148 | 1480 | 5920 | 376 |
| 4 | 178 | 1780 | 7120 | 66 |
| 5 | 178 | 1780 | 7120 | 66 |
| 6 | 178 | 1780 | 7120 | 66 |

The robot toolbox [17] in MATLAB was used to model aubo-i5, as shown in Figure 2.

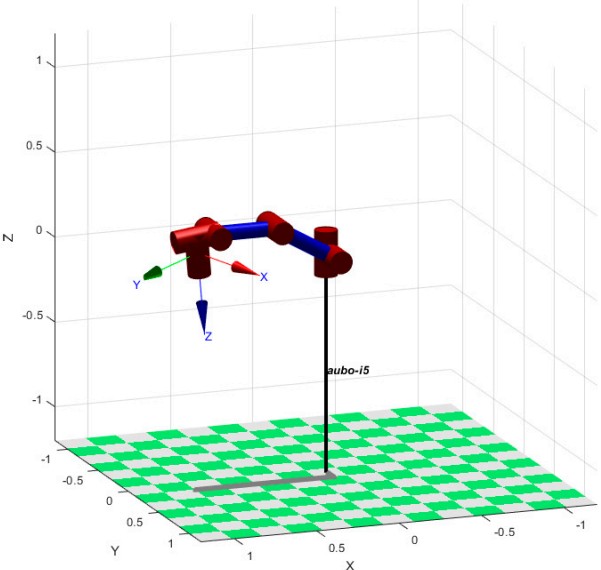

**Figure 2.** Robot model in MATLAB.

## 2.2. Forward Kinematics Analysis

Forward kinematics is to obtain the end-effector pose relative to the base using the geometric parameters and joint coordinates of the robot. According to the relative relation between link *i* and link *i* + 1 of the robot, its relative pose is expressed by 4 × 4 transformation matrix. Using the improved DH parameters in Table 1, the transformation matrix of each link can be obtained as follows,

$$
{}^0_1T = \begin{bmatrix} c_1 & -s_1 & 0 & 0 \\ s_1 & c_1 & 0 & 0 \\ 0 & 0 & 1 & d_1 \\ 0 & 0 & 0 & 1 \end{bmatrix}
{}^1_2T = \begin{bmatrix} c_2 & -s_2 & 0 & 0 \\ 0 & 0 & 1 & d_2 \\ -s_2 & -c_2 & 0 & 0 \\ 0 & 0 & 0 & 1 \end{bmatrix}
{}^2_3T = \begin{bmatrix} c_3 & -s_3 & 0 & a_2 \\ -s_3 & -c_3 & 0 & 0 \\ 0 & 0 & -1 & -d_3 \\ 0 & 0 & 0 & 1 \end{bmatrix}
$$
$$
{}^3_4T = \begin{bmatrix} c_4 & -s_4 & 0 & a_3 \\ -s_1 & -c_4 & 0 & 0 \\ 0 & 0 & -1 & -d_4 \\ 0 & 0 & 0 & 1 \end{bmatrix}
{}^4_5T = \begin{bmatrix} c_5 & -s_5 & 0 & 0 \\ 0 & 0 & 1 & d_5 \\ -s_5 & -c_5 & 0 & 0 \\ 0 & 0 & 0 & 1 \end{bmatrix}
{}^5_6T = \begin{bmatrix} c_6 & -s_6 & 0 & 0 \\ 0 & 0 & -1 & -d_6 \\ s_6 & c_6 & 0 & 0 \\ 0 & 0 & 0 & 1 \end{bmatrix}
\tag{1}
$$

where ${}^i_{i+1}T$ is the transformation matrix between adjacent links, $s_i$ is $sin\theta_i$, and $c_i$ is $cos\theta_i$.

Multiply each link matrices of aubo-i5 as;

$$
{}^0_6T = {}^0_1T(\theta_1) \, {}^1_2T(\theta_2) \, {}^2_3T(\theta_3) \, {}^3_4T(\theta_4) \, {}^4_5T(\theta_5) \, {}^5_6T(\theta_6)
\tag{2}
$$

and the forward kinematics equation is given by:

$$
{}^0_6T = \begin{bmatrix} r_{11} & r_{12} & r_{13} & p_x \\ r_{21} & r_{22} & r_{23} & p_y \\ r_{31} & r_{32} & r_{33} & p_z \\ 0 & 0 & 0 & 1 \end{bmatrix}
\tag{3}
$$

where

$$
r_{11} = c_1[(c_2c_3 + s_2s_3)(c_4c_5c_6 - s_4s_6) - (s_4c_5c_6 + c_4s_6)(s_2c_3 - c_2s_3)] + s_1s_5c_6
\tag{4}
$$

$$
r_{12} = -c_1[(c_2c_3 + s_2s_3)(c_4c_5c_6 + s_4c_6) - (s_2c_3 - c_2s_3)(s_4c_5c_6 - c_4c_6)] - s_1s_5s_6
\tag{5}
$$

$$
r_{13} = c_1[(c_2c_3 + s_2s_3)c_4s_5 - s_4s_5(s_2c_3 - c_2s_3) - s_1c_5
\tag{6}
$$

$$r_{21} = s_1[(c_2c_3 + s_2s_3)(c_4c_5c_6 - s_4s_6) - (s_4c_5c_6 + c_4s_6)(s_2c_3 - c_2s_3)] - c_1s_5c_6 \tag{7}$$

$$r_{22} = -s_1[(c_2c_3 + s_2s_3)(c_4c_5c_6 + s_4c_6) - (s_2c_3 - c_2s_3)(s_4c_5c_6 - c_4c_6)] + c_1s_5s_6 \tag{8}$$

$$r_{23} = s_1[(c_2c_3 + s_2s_3)c_4s_5 - s_4s_5(s_2c_3 - c_2s_3) + c_1c_5 \tag{9}$$

$$r_{31} = (c_2s_3 - s_2c_3)(c_4c_5c_6 - s_4s_6) - (s_2s_3 + c_2c_3)(s_4c_5c_6 - c_4s_6) \tag{10}$$

$$r_{32} = -(c_2s_3 - s_2c_3)(c_4c_5s_6 + s_4c_6) + (s_2s_3 + c_2c_3)(s_4c_5c_6 - c_4c_6) \tag{11}$$

$$r_{33} = (c_2s_3 - s_2c_3)c_4s_5 - s_4s_5(s_2s_3 + c_2c_3) \tag{12}$$

$$p_x = c_1[(c_2c_3 + s_2s_3)(c_4s_5d_6 - s_4d_5 + a_3) - (s_2c_3 - c_2s_3)(s_4s_5d_6 - c_4d_5)] + c_2a_2] - s_1(c_5d_6 + d_4 + d_2 - d_3) \tag{13}$$

$$p_y = s_1[(c_2c_3 + s_2s_3)(c_4s_5d_6 - s_4d_5 + a_3) - (s_2c_3 - c_2s_3)(s_4s_5d_6 - c_4d_5)] + c_2a_2] + c_1(c_5d_6 + d_4 + d_2 - d_3) \tag{14}$$

$$p_z = (c_2s_3 - s_2c_3)(c_4s_5d_6 - s_4d_5 + a_3) - (s_2s_3 + c_2c_3)(s_4s_6d_6 + c_4d_5) - s_2a_2 + d_1 \tag{15}$$

*2.3. Inverse Kinematic Analysis*

Inverse kinematics is to obtain the joint coordinates using the geometric parameters and end-effector pose relative to the base of the robot [18]. In this paper, the method in [19] is used to multiply and invert the transformation matrix above to calculate the joint coordinates as:

$$\theta_1 = \pm\arctan\left(\frac{d_6}{\sqrt{(r_{13}d_6 - p_x)^2 + (p_y - r_{23}d_6)^2 - d_2^2}}\right) - \arctan\left(\frac{p_y - r_{23}d_6}{r_{13}d_6 - p_x}\right) \tag{16}$$

$$\theta_2 = \arctan\{[Xs_3a_3 + Y(a_2 + a_3c_3)] / [X(a_2 + a_3c_3) - Ys_3a_3]\} \tag{17}$$

$$\theta_3 = \pm\arccos\left[(X^2 + Y^2 - a_3^2 - a_2^2) / 2a_2a_3\right] \tag{18}$$

$$\theta_4 = \arctan[(c_1s_{23}r_{13} - s_1s_{23}c_1 - c_{23}r_{33}) / (c_1c_{23}r_{23} + s_1c_{23}r_{23} - s_{23}r_{33})] \tag{19}$$

$$\theta_5 = \pm\arccos(-s_1r_{13} + c_1r_{23}) \tag{20}$$

$$\theta_6 = \arctan[(s_1r_{11} - c_1r_{12}) / (c_1r_{22} - s_1r_{21})] \tag{21}$$

where $s_{23}$ is $\sin(\theta_2 - \theta_3)$, $c_{23}$ is $\cos(\theta_2 - \theta_3)$ and X, Y are:

$$X = -d_5[s_6(c_1r_{11} + s_1r_{21}) + c_6(c_1r_{12} + s_1r_{22})] - d_6(c_1r_{13} + s_1r_{23}) + c_1p_x + s_1p_y \tag{22}$$

$$Y = -d_5(s_2r_{31} + c_5r_{32}) - d_6r_{33} + p_z - d_1 \tag{23}$$

Both $\theta_3$ and $\theta_5$ have positive and negative solutions, indicating that there are different joint coordinates for the same end-effector pose.

## 3. Multi-Objective Trajectory Planning

*3.1. Problem Statement*

From the initial pose to the final pose, the trajectory sequence $T_i$ of the end-effector is obtained. Use the above inverse kinematics method to calculate the position-time sequence as:

$$Q = \left\{(q_{ij}, t_i) | i = 0, 1 \ldots, n, j = 1, 2 \ldots, N\right\} \tag{24}$$

where $n + 1$ is the number of time nodes, $N$ is the number of joints, $q_{ij}$ is the position of joint j and time $i$.

In addition, in order to make the movement of each trajectory position smooth transition, a specific curve is used for interpolation. Construct a trajectory function with time on the horizontal axis and position on the vertical axis.

On the premise of satisfying the performance constraints, the trajectory function is optimized from three aspects: operating efficiency, energy consumption, and operating smoothness. For these three aspects, three optimization goals need to be defined, which can be considered as following:

1.  Improving the operating efficiency of the robot can be achieved by reducing the overall operating time;
2.  Reducing energy consumption of the robot can be achieved by reducing the average acceleration of each joint. Because the average acceleration is related to the torque, which is related to the energy consumption of the motor;
3.  Improving the smoothness of the robot can be achieved by reducing the average jerk, because the average jerk represents the speed of torque change. A smaller jerk can make the torque change smoother.

Therefore, we define three optimization goals: total motion time S1, average acceleration S2, and average jerk S3, which respectively correspond to the optimization in the above three aspects. Meanwhile, define each time interval $\Delta t_i$ as the optimization variable, and the maximum velocity, $v_{jmax}$, acceleration $a_{jmax}$ and jerk $j_{jmax}$ of each joint as constraints;

$$S1 = \sum_{i=1}^{n} \Delta t_i = \mathrm{T} \tag{25}$$

$$S2 = \sum_{m=1}^{M} \sqrt{\frac{1}{T} \int_{0}^{T} a_i^2 dt} \tag{26}$$

$$S3 = \sum_{m=1}^{M} \sqrt{\frac{1}{T} \int_{0}^{T} j^2 dt} \tag{27}$$

$$\begin{cases} g_1 = \left|v_{ij}\right| - v_{jmax} \le 0 \\ g_2 = \left|a_{ij}\right| - a_{jmax} \le 0 \\ g_3 = \left|j_{ij}\right| - j_{jmax} \le 0 \end{cases} \tag{28}$$

where $v_{ij}$, $a_{ij}$, $j_{ij}$ are respectively the velocity, acceleration and jerk of joint j and time *i*.

For the above multi-objective optimization problem, the three optimization goals are contradictory, and it is impossible to achieve the optimal simultaneously. Generally speaking, the multi-objective optimization algorithm can be used to obtain an optimal solution set instead of an optimal solution, and the solutions in the solution set cannot be compared with each other [20]. These solutions, while improving any objective function, necessarily weaken at least one other objective function. These solutions are called Pareto solutions, and the set of optimal solutions for a set of objective functions is called Pareto optimal solution sets [21].

In this paper, the multi-objective optimization algorithm is used to solve the multi-objective optimization problem satisfying the constraint conditions, and Pareto optimal solution set of joint trajectories is generated.

### 3.2. Joint Trajectory Constructed Based on B-Spline

The B-spline curve [22] is used to construct the joint trajectory of the manipulator, which is uniformly described as:

$$p(u) = \sum_{i=0}^{n} d_i N_{i,k}(u) \tag{29}$$

where $d_i$ is the control vertex, $N_{i,k}(u)$ is the k order standard B−spline basis function, $u = [u_0, u_1, \ldots, u_{n+2k}]$ is the node vector, and $p(u)$ is the joint position at node u.

In order to interpolate the trajectory, the segment connection points of the trajectory curve correspond to the nodes in the B-spline definition domain, and the first and last points are consistent

with the first and last data points. Using the accumulative chord length parameterization method to normalize the time node $t_i$ to get the inner node value;

$$u_i = u_{i-1} + |\Delta t_{i-k-1}| / \sum_{j=0}^{n-1} \Delta t_j, \ i = k+1, \ldots, n+k-1 \tag{30}$$

where $\Delta t_k = t_{k+1} - t_k \ (k = 0, 1, \ldots, n-1)$ is the chord length.

The $n + 1$ equation satisfying the interpolation condition is given by:

$$p(u_{i+k}) = \sum_{j=i}^{i+k} d_j N_{j,k}(u_{i+k}) = P_i \tag{31}$$

where $u_{i+k} \in [u_k, u_{n+k}], i = 0,1,...,n$.

Therefore, $k-1$ additional conditions are required to invert the control vertex $d_i$. For the seven-order B-spline curve, 6 tangent boundary conditions are added:

$$\begin{cases} p'(u)\big|_{u=u_7} = v_s, & p'(u)\big|_{u=u_{n+7}} = v_e \\ p''(u)\big|_{u=u_7} = a_s, & p''(u)\big|_{u=u_{n+7}} = a_e \\ p'''(u)\big|_{u=u_7} = j_s, & p'''(u)\big|_{u=u_{n+7}} = j_e \end{cases} \tag{32}$$

where $p'(u)$, $p''(u)$, $p'''(u)$ are respectively 1, 2, 3 derivative vectors of B−spline. $v_s$, $v_e$, $a_s$, $a_e$, $j_s$, $j_e$ are respectively the velocity, acceleration and jerk at the first and last point of the curve trajectory. According to the requirements of robot trajectory planning, the six parameters are defined as 0.

The above derivative vectors can be expressed as:

$$p^r(u) = \sum_{j=i-k}^{i-r} d_j^r N_{j,k-r}(u), u \in [u_i, u_{i+1}] \tag{33}$$

where the $d_j^r$ is given by

$$d_j^l = \begin{cases} d_j, \ l = 0 \\ (k-l+1)\dfrac{d_{j+1}^{l-1} - d_j^{l-1}}{u_{j+k+1-l} - u_j}, \ l = 1, 2, \ldots, r \\ j = i-k, \ i-k+1, \ldots, i-r \end{cases} \tag{34}$$

In addition, the B-spline curve has a convex hull property, so the constraints of the B-spline control vertex are equal to the maximum velocity $v_{jmax}$, acceleration $a_{jmax}$, and jerk $j_{jmax}$ of each joint of the robot:

$$\begin{cases} \max\left\{\left|d_{ij}^1\right|\right\} \le v_{jmax}, \ i = 1, 2, \ldots, n+k-1 \\ \max\left\{\left|d_{ij}^2\right|\right\} \le a_{jmax}, \ i = 1, 2, \ldots, n+k-1 \\ \max\left\{\left|d_{ij}^3\right|\right\} \le j_{jmax}, \ i = 1, 2, \ldots, n+k-1 \end{cases} \tag{35}$$

where $d_{ij}^1$, $d_{ij}^2$ and $d_{ij}^3$ are respectively the control vertex of the joint j and point i in the velocity, acceleration and jerk curve.

According to the above process, the control vertices are obtained, and then the trajectory interpolation of the B-spline is completed.

### 3.3. Trajectory Competitive Multi-Objective Particle Swarm Optimization

Multi-objective particle swarm optimization (MOPSO) [23] is a swarm intelligence algorithm, which simulates the foraging behavior of birds to search the solution space. The algorithm has the advantages of simple structure and fast convergence speed. The basic idea is that there are N particles forming a population in the D-dimensional search space, wherein the particle i is $X_i = (x_{i1}, x_{i2}, \ldots, x_{iD})^T$, which represents a potential solution of the problem. The velocity of the particle i is $V_i = (V_{i1}, V_{i2}, \ldots, V_{iD})^T$, the individual extreme value is $P_i = (P_{i1}, P_{i2}, \ldots, P_{iD})^T$, and the population

extreme value of the population is $P_g = (P_{g1}, P_{g2}, \ldots, P_{gD})^T$. Particles update their own speed and position through individual extreme values and group extreme values:

$$\begin{cases} V_{id}^{k+1} = \omega V_{id}^k + c_1 r_1 (P_{id}^k - X_{id}^k) + c_2 r_2 (P_{gd}^k - X_{gd}^k) \\ \qquad\qquad V_{id}^{k+1} = V_{id}^k + V_{k+1\,id} \\ \qquad\qquad d = 1, 2, \ldots, D, i = 1, 2, \ldots, n \end{cases} \tag{36}$$

where $\omega$ is the inertia weight, k is the current iteration number, $V_{id}$ is the particle speed, $c_1$ and $c_2$ are the acceleration factors, $r_1$ and $r_2$ are random numbers between (0, 1).

However, when dealing with complex optimization problems, due to the characteristics of fast convergence, the algorithm is easy to fall into the local optimal and the diversity is poor. In order to solve the above problems, Zhang [24] proposed a competition multi-objective particle swarm optimization (CMOPSO), which adopted the elite strategy in NSGA-II [25] algorithm to update the particles. The algorithm flow is shown in Figure 3.

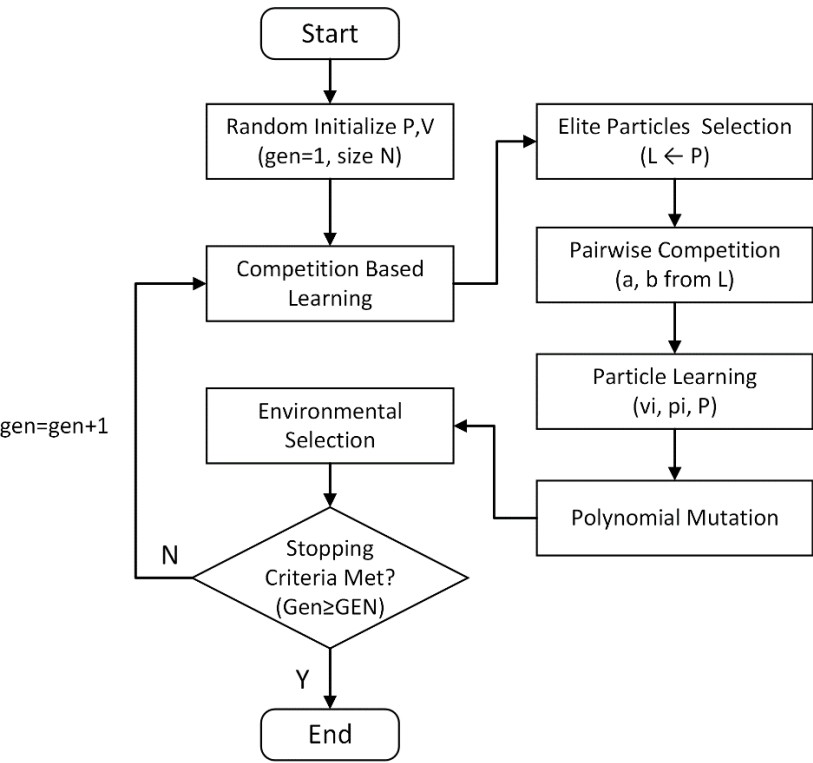

**Figure 3.** Competition multi-objective particle swarm optimization (CMOPSO) algorithm optimization process.

This paper uses this algorithm to optimize the trajectory. However, the algorithm is an unconstrained search mechanism, which needs to be improved by adding constraints when applied to constrained multi-objective optimization problems. In this paper, the feasibility rules in NSGA-II [25] are used to calculate the constraint violation of particle x in constraint j:

$$G_j = \begin{cases} \max\{g_j(x), 0\}, \ 1 \leq j \leq q \\ \max\{|h_j(x)| - \delta, 0\}, \ q+1 \leq j \leq m \end{cases} \tag{37}$$

where $g_j(x)$ is the inequality constraint, $h_j(x)$ is the equality constraint, and $\delta$ is the tolerance parameter of the equality constraint, which is defined as a specific value (generally set to 0.0001). The total constraint violation of particle x is:

$$v(x) = \sum_{j=1}^{m} G_j(x) \tag{38}$$

According to the constraint violation of particle x, the non-dominated ranking with feasibility rules is adopted in the step of elite particles selection. Determine the dominance relationship of the two particles according to the following feasibility rules:

1.  $x_i$ is feasible solution, $x_j$ is infeasible solution, then choose $x_i$;
2.  $x_i$ and $x_j$ are both feasible solutions, $x_i$ has better objective function, then choose $x_i$;
3.  $x_i$ and $x_j$ are both infeasible solutions, $x_i$ has smaller constraint violation, then choose $x_i$.

In this paper, CMOPSO is combined with the feasibility rules and kinematic constraints to obtain the trajectory competition multi-objective particle swarm optimization (TCMOPSO). The algorithm can be widely used in robot multi-objective trajectory optimization. Taking the time interval of the trajectory sequence as the decision variable, Equations (25)–(27) as the optimization objectives, and Equation (28) as the constraint conditions, the basic process of using TCMOPSO for trajectory optimization is as follows:

(1)  Parameter setting: trajectory-time sequence and kinematic constraints of each joint are known; set the range of decision variables (the minimum value is obtained according to known conditions, the maximum value is user-defined); set the population size $N$, the maximum iterations number GEN.
(2)  Initialization: randomly generate particle position set P and particle velocity set $V$.
(3)  Competitive learning: The elite particle set $L$ is obtained through non-dominated ranking. For each particle $p$ in $P$, pick any two particles $a$ and $b$ in L to match and compete, then update the position and speed of particle p according to the winner. Add the updated $p$ to $P'$, and perform polynomial mutation on $P'$.
(4)  Natural selection: Natural selections for $P$ and $P'$ to get a new $P$.
(5)  Repeat (3) and (4) until the number of iterations gen reaches the maximum iterations number GEN.

Through the above steps, Pareto optimal solution satisfying kinematic constraints is obtained, and the multi-objective trajectory optimization of the robot is completed.

## 4. Experiment

### 4.1. Trajectory Interpolation

In order to verify the effectiveness of the proposed multi-objective trajectory planning method, the collaborative robot AUBO-I5 is used for the experiment. Firstly, define a space trajectory arbitrarily, and use inverse kinematics algorithm to convert it into joint space trajectory, as shown in Table 3.

**Table 3.** Robot joint space trajectory.

| Node | Joint 1 (°) | Joint 2 (°) | Joint 3 (°) | Joint 4 (°) | Joint 5 (°) | Joint 6 (°) |
|---|---|---|---|---|---|---|
| 1 | 16.99 | −33.12 | 43.89 | 25.70 | 110.36 | −25.95 |
| 2 | 17.23 | −13.88 | 47.70 | 60.69 | 105.23 | −28.51 |
| 3 | 17.88 | −19.60 | 55.23 | 79.05 | 103.25 | −16.05 |
| 4 | 14.39 | −28.46 | 63.88 | 75.21 | 97.78 | −5.21 |
| 5 | 11.25 | −41.72 | 68.35 | 72.70 | 86.04 | 6.50 |
| 6 | 3.10 | −45.34 | 72.41 | 67.45 | 79.94 | 12.20 |
| 7 | −20.13 | −42.69 | 79.65 | 53.54 | 72.76 | 16.24 |
| 8 | −25.78 | −40.34 | 82.17 | 47.80 | 78.16 | 13.55 |

After obtaining the robot trajectory in Table 3, the interpolation function is used to construct the time-joint sequence. The trajectory to be planned by the collaborative robot is relatively complex, and it requires higher accuracy and stability. It is necessary to select an interpolation function suitable for the collaborative robot. In recent years, some researchers have used B-spline functions to interpolate robots, for example, quintic B-spline functions are used to interpolate trajectories [26]. However, this method has the disadvantage that the jerk value at the beginning and the end cannot be specified. For collaborative robots, the unsmooth jerk value may cause a sudden change in torque, which threatens the safety of the collaborators.

Select the trajectory of the robot joint 4, use cubic spline curve [7], quintic B-spline curve [26] and seven-order B-spline curve to construct the trajectory curve. Compare the jerk curves obtained by the three methods, as shown in Figure 4.

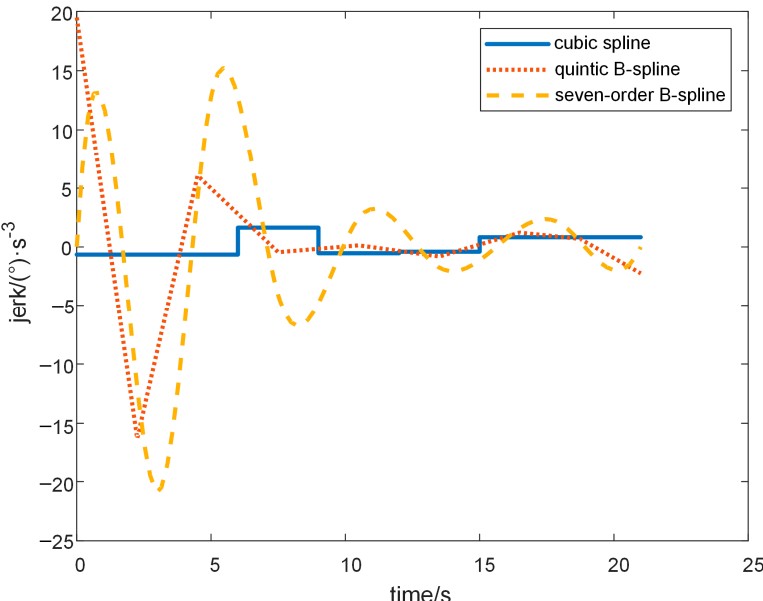

**Figure 4.** The comparison of three spline method.

It can be seen from Figure 4 that the jerk curve of the cubic spline is not continuous. The jerk curve of the quintic B-spline is not smooth and the value at the beginning and the end are not 0. The operation time of the three methods in MATLAB is compared, as shown in Table 4.

**Table 4.** The operation time of the three methods.

| Spline | Cubic Spline | Quintic B-Spline | Seven-Order B-Spline |
| --- | --- | --- | --- |
| Time (ms) | 2.884 | 3.734 | 4.232 |

The calculation times of the three methods in MATLAB are all small enough, so it is no computational burden to choose high-order B-spline for trajectory interpolating. In this paper, the seven-order B-spline function is used to set the first and last trajectories, acceleration, and jerk to 0, to ensure the continuity of speed, acceleration, and jerk. In this paper, the seven-order B-spline function is used to construct the trajectory equation, and the velocity, acceleration and jerk value of the first and last are all set to 0.

### 4.2. Trajectory Optimization

After constructing the joint trajectory curve, the time interval of the trajectory is used as the decision variable, and the optimal variable value is selected by the optimization algorithm. The

time interval of the joint sequence is used as the decision variable. The upper bound of the decision variable is six, and the lower bound is calculated. The population size is set to 100 and the maximum evolutionary algebra is 10,000 generations.

Figure 5 shows the Pareto front obtained by the algorithm proposed in this paper. The solution closer to A has the better energy and jerk performance, and the worse time performance. In contrast, the solution closer to B has the better time performance, and the worse energy and jerk performance. Therefore, energy is positively correlated with jerk performance and negatively correlated with time performance.

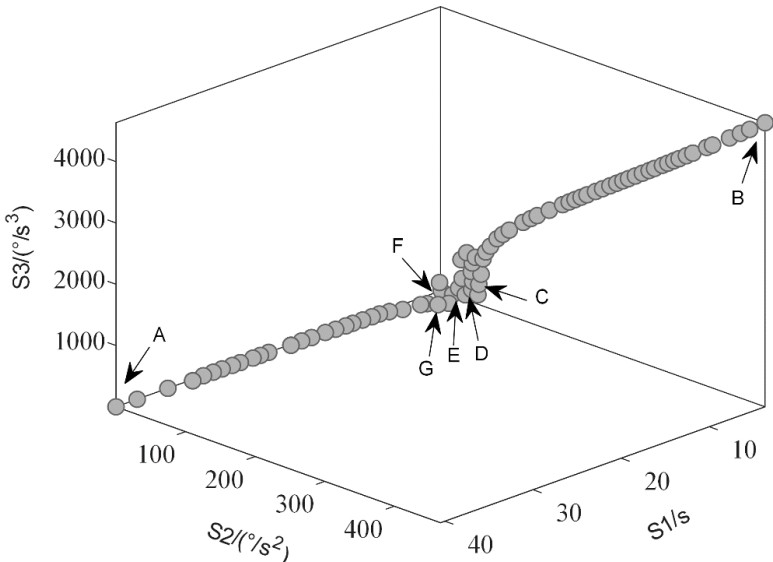

**Figure 5.** Pareto front of the proposed algorithm.

The objective function of each point is shown in Table 5. Both A and B are close to the boundary of the constraint. The optimal solution obtained by the algorithm proposed in this paper covers the whole solution space well.

**Table 5.** Comparison of trajectory optimization performance.

| Result | S1/s | S2/((°)/$s^2$) | S3/((°)/$s^3$) |
| --- | --- | --- | --- |
| A | 41.5781 | 0.1027 | 0.1158 |
| B | 3.2637 | 468.6523 | 4728.5563 |

Several popular multi-objective optimization algorithms are added with the kinematic constraints of the collaborative robot. Figure 6 shows the Pareto front obtained by these multi-objective optimization algorithms.

For the trajectory optimization problem, the solution set obtained by MOPSO [23] has some disconnections. The solution set obtained by NSGAII [25] and SPEA2 [27] does not cover the entire solution space. The range of solution set obtained by MOEA/D [28] is relatively small. Compared with these algorithms, the solution set obtained by the algorithm proposed in this paper is more uniformly distributed and has better continuity.

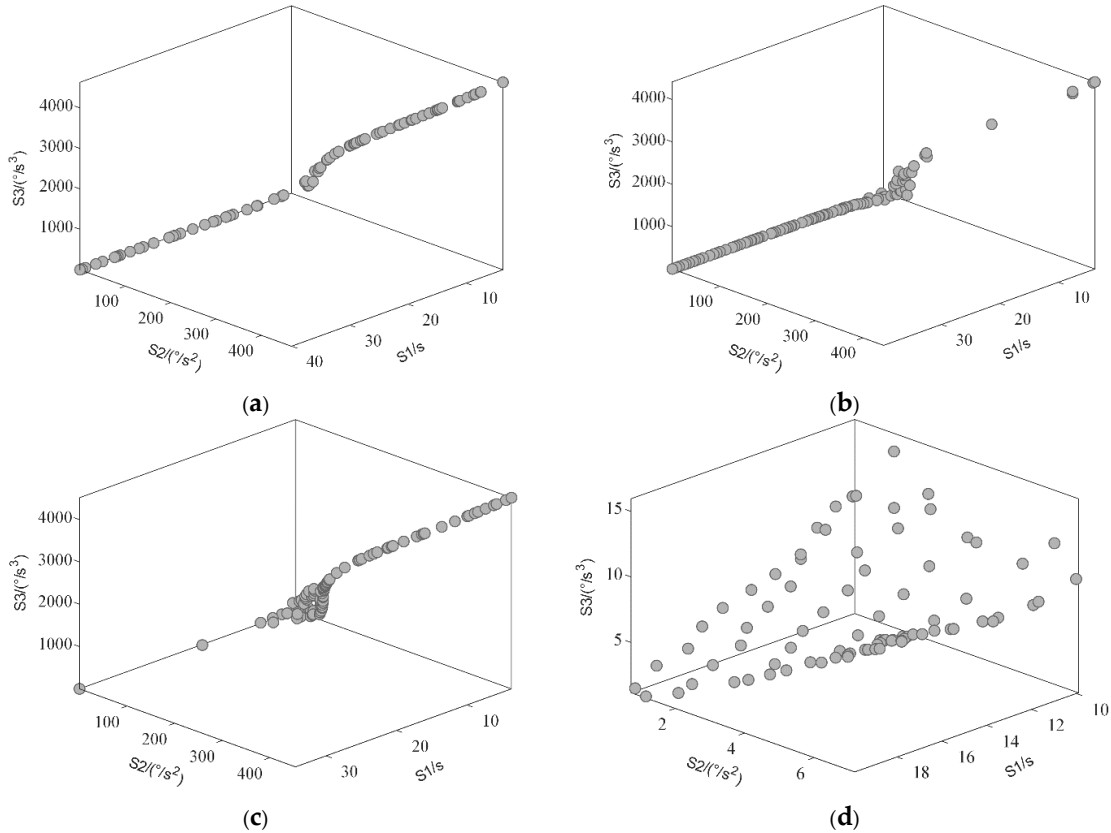

**Figure 6.** Several multi-objective optimization algorithms are used for trajectory optimization, where (**a**) multi-objective particle swarm optimization (MOPSO), (**b**) nondominated sorting genetic algorithm II (NSGAII), (**c**) Strength Pareto Evolutionary Algorithm 2 (SPEA2), and (**d**) Multi-objective evolutionary algorithm based on decomposition (MOEA/D).

In the actual robot operation, three objectives should be considered comprehensively to obtain the required solution. For some specific scenarios and tasks, such as human-robot collaboration, the maximum velocity, acceleration and jerk are required to set in particular values to meet the actual needs. The optimal objective values need to satisfy the specified maximum range simultaneously.

This paper defines the normalized weight function and sets the weight coefficients for the three objectives.

$$f = \sigma_1 \frac{S1 - S1_{min}}{S1_{max} - S1_{min}} + \sigma_2 \frac{S2 - S2_{min}}{S2_{max} - S2_{min}} + \sigma_3 \frac{S3 - S3_{min}}{S3_{max} - S3_{min}} \tag{39}$$

where $\sigma_1$, $\sigma_2$ and $\sigma_3$ are the weight coefficients of each objective function, and $S1_{max}$ and $S1_{min}$ are the maximum and minimum values of objective S1. The function maps the three objectives values to the interval [0,1], and then multiplies the customized weight coefficient to obtain the actual optimal solution. In brief, if higher efficiency is required, give $\sigma_1$ a larger value. If a lower energy consumption and a higher smoothness are required, give $\sigma_2$ and $\sigma_3$ larger values. This paper conducts experiments on several sets of typical weight coefficients, as shown in Table 6. Further, the results C to G are shown in Figure 5.

**Table 6.** Comparison of different weight coefficients.

| Result | $\sigma_1$ | $\sigma_2$ | $\sigma_3$ | S1/s | S2/((°)/$S^2$) | S3/((°)/$S^3$) |
|--------|-----------|-----------|-----------|---------|----------------|----------------|
| C | 10 | 1 | 1 | 4.0497 | 63.6947 | 466.5722 |
| D | 5 | 1 | 1 | 4.4629 | 51.5607 | 205.0956 |
| E | 1 | 1 | 1 | 5.8017 | 16.8444 | 195.3287 |
| F | 1 | 5 | 5 | 9.8286 | 8.0456 | 36.1065 |
| G | 1 | 10 | 10 | 11.4246 | 2.5995 | 15.8354 |

In some human-robot collaboration scenarios, the safety of collaborators must be carefully considered. Under the premise of ensuring the operation efficiency, more consideration is given to the stability and smoothness to ensure the smooth progress of human-robot collaboration. Therefore, the solution at point F is adopted as the optimal solution in this paper. The optimal objective value is [9.8286, 8.0456, 36.1065], and the optimal time node is [0, 1.7779, 2.9080, 4.7470, 5.9863, 7.0328, 8.5141, 9.8286]. The trajectory-time curves of each joints are shown in Figure 7.

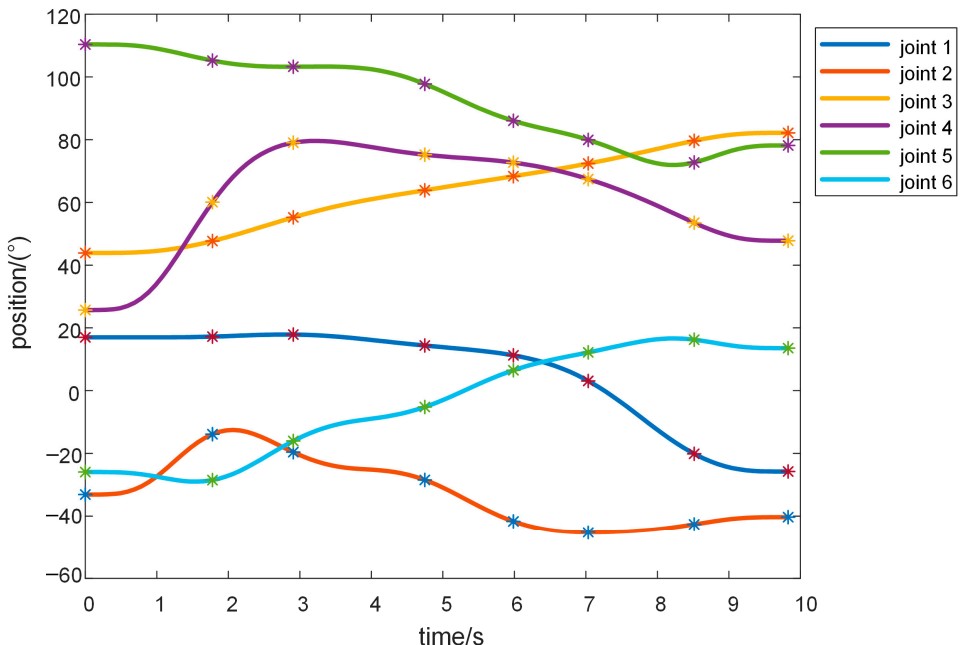

**Figure 7.** Joint trajectory-time sequence.

The velocity, acceleration and jerk curves of each joints are shown in Figure 8. The curves are continuous and smooth, with no shocks or sudden changes. The range of the maximum and minimum values are consistent with the kinematic constraints of the robot. In addition, the value of velocity, acceleration and jerk at the beginning and the end are all 0. It will not cause a sudden change of torque, which meets the needs of collaborative robot operation.

For collaborative robots, the operation safety of collaborators during operation is particularly important. The trajectory constructed by the algorithm proposed in this paper makes the robot running process more efficient and stable. The possibility of abnormal situations is reduced, and the safety of collaborators is well guaranteed. In addition, the robot's reducer, motor and other components are well protected, which is beneficial to extend the service life.

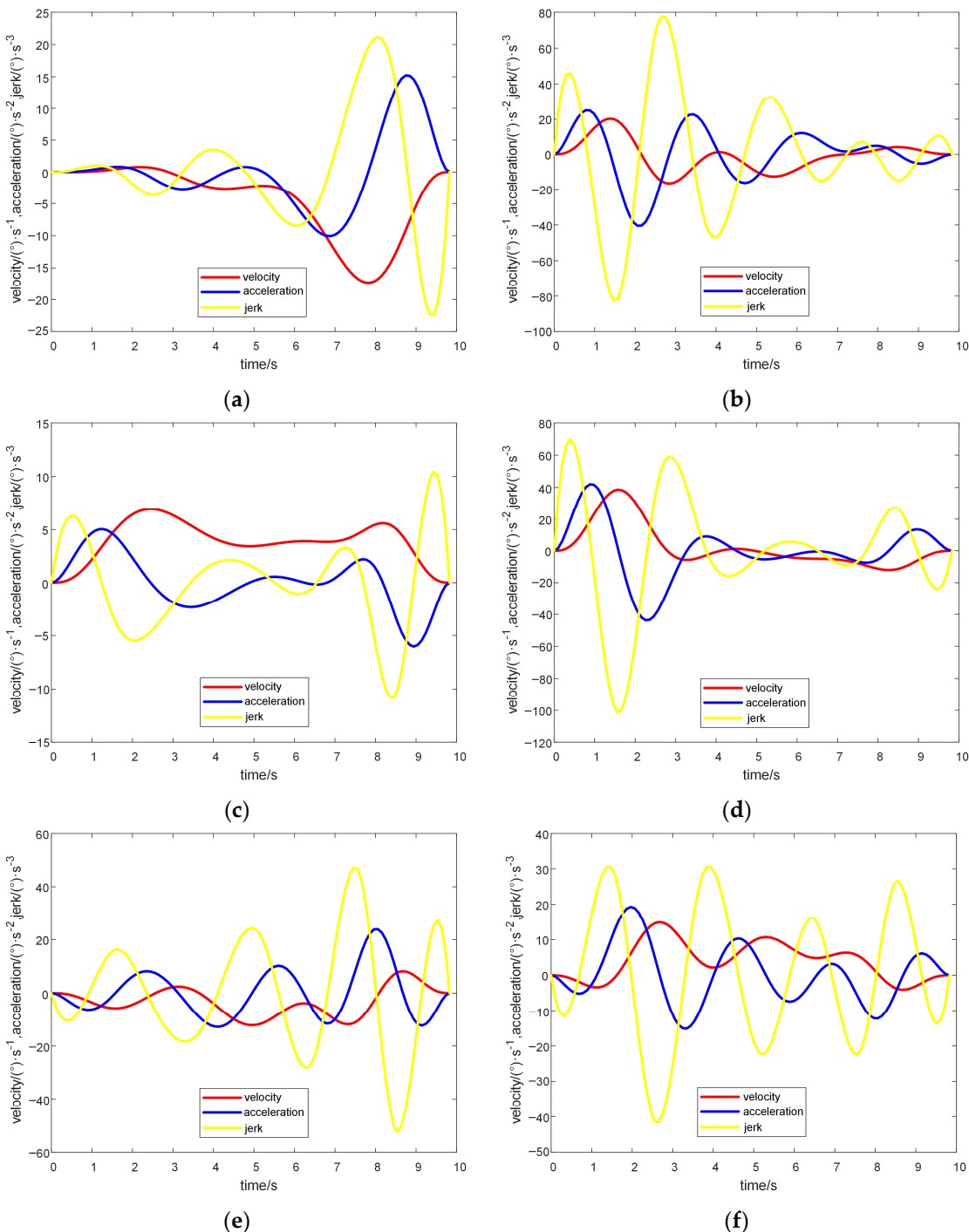

**Figure 8.** Velocity, acceleration and jerk curves of each joint, where Figure (**a**–**f**) represent Joint 1 to Joint 6 respectively.

## 4.3. Collaborative Robot Control

The ROS system is a system framework developed for robots, based on the Ubuntu system environment [29]. The ROS system uses the TCP/IP to connect to the controller of the collaborative robot, and transmits trajectory planning data in real time [30]. MoveIt is a robot simulation control software in the ROS system, which can build a robot model and make motion planning for the robot. Establish the kinematics model of the AUBO-I5 in MoveIt, as shown in Figure 9.

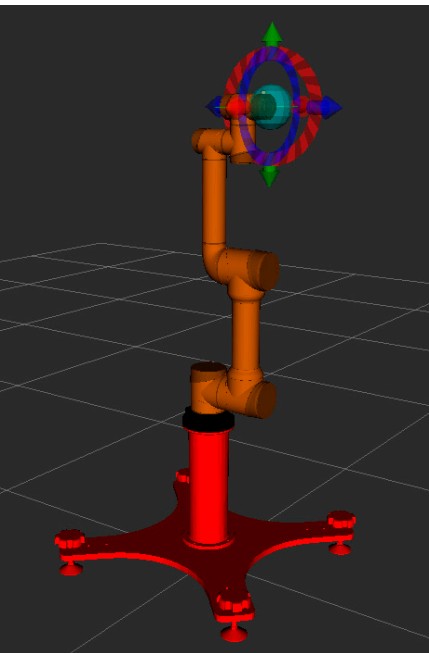

**Figure 9.** Collaborative robot model in MoveIt.

In order to quickly generate Cartesian space trajectories for collaborative robots, a trajectory editor software was designed. As shown in Figure 10. The editor can input end-effector position and pose to insert trajectory points. The trajectory points are listed in the table, and the generated trajectory list can be edited. In addition, the editor displays the designed trajectory in three dimensions for easy adjustment. The trajectory editing tool is based on a web page. The interface is simple, and it can be used across platforms to meet various scenarios of collaborative robot operation.

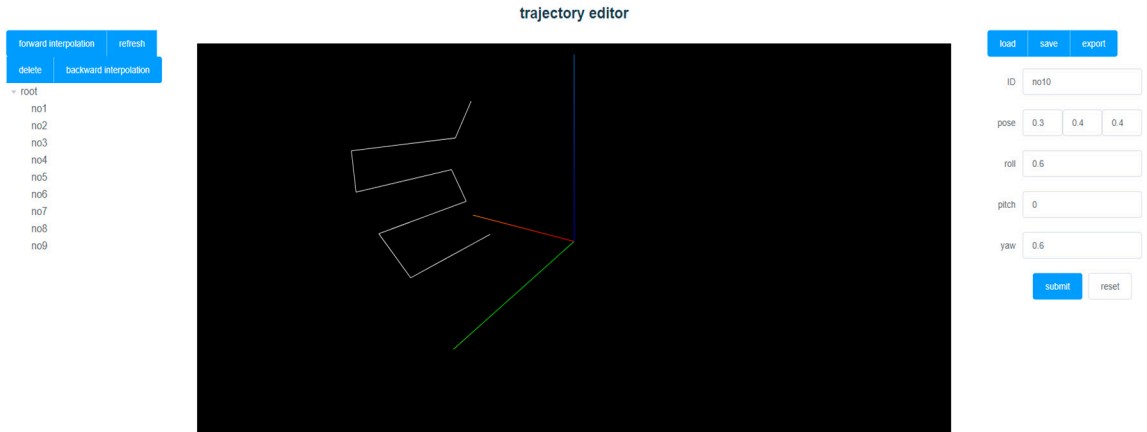

**Figure 10.** Editor interface.

The joint trajectory information is transferred to the ROS system, and ROS node information is released. Use MoveIt to load the trajectory time node, position, speed, acceleration and other information to control the collaborative robot in real time. The experiment proves that the collaborative robot runs smoothly without vibration or other conditions, as shown in Figure 11.

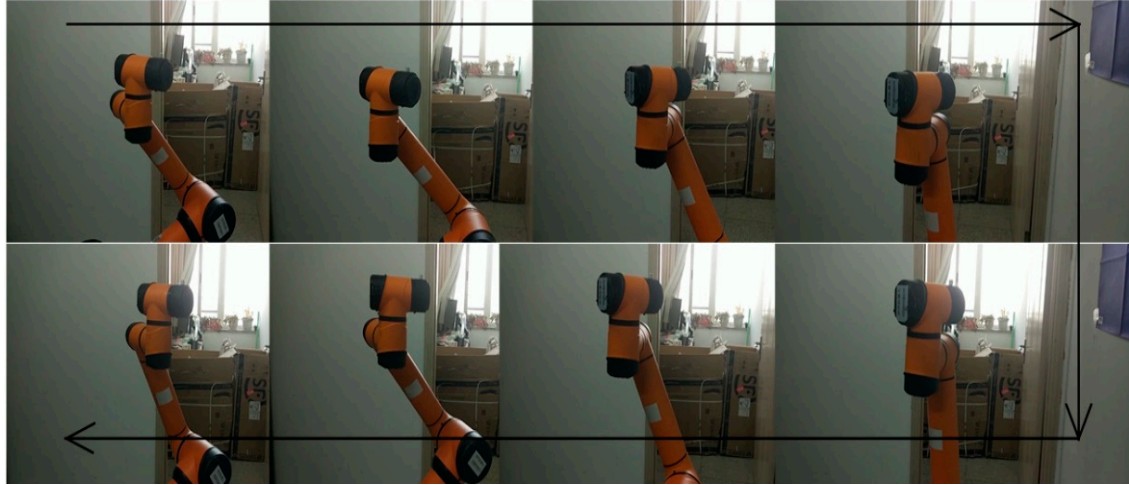

**Figure 11.** Collaborative robot operation.

## 5. Conclusions

This paper proposed a trajectory planning method for collaborative robots. The seven order B-spline is utilized to construct a continuous trajectory to ensure the continuous position, velocity, acceleration and jerk of each joint. Then, we proposed a trajectory competitive multi-objective particle swarm optimization (TCMOPSO) algorithm to optimize the joint trajectory to obtain the Pareto optimal solution set. For different scenarios and tasks, we proposed the normalized weight function, which can be used to select the most suitable solution from the solution set. Finally, we design a trajectory editor software to generate Cartesian space trajectories and use the method proposed in this paper to conduct actual robot control experiments. The experimental results show that the obtained trajectory is smooth and controllable. The algorithm proposed in this paper can be widely used in various collaboration robots, and provide ideas for the study of trajectory planning algorithms.

In future work, we are planning to improve the method proposed in this paper so that it can be used in more scenarios and tasks. In addition, we will continue to research the motion planning of collaborative robots, include how to avoid obstacles and collaborate with humans safely. This paper does not consider the dynamics and robot pose accuracy, which are very important in human-robot collaboration. We need to control the robot torque precisely to ensure the smooth completed of the task and the safety of human beings.

**Author Contributions:** Conceptualization, J.L. and Y.X.; methodology, J.L.; software, J.L.; validation, J.L. and Y.X.; formal analysis, J.L.; writing—original draft preparation, J.L.; writing—review and editing, J.L., Y.X., M.C. and G.L.; supervision, Y.X.; project administration, J.L. All authors have read and agreed to the published version of the manuscript.

**Funding:** This work is supported by Beijing Natural Science Foundation (Grant No.4192023 and 4202024); Natural Science Foundation of China (Grant NO.61603047); the Qin Xin Talents Cultivation Program of Beijing Information Science and Technology University (Grant No. QXTCPC201704).

**Conflicts of Interest:** The authors declare no conflicts of interest.

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
