# Peer review of "A Multi-Objective Trajectory Planning Method for Collaborative Robot"

_electronics, doi:10.3390/electronics9050859_

Round 1
Reviewer 1 Report
After reviewing the paper entitled A multi-objective trajectory planning method for collaborative robot, I have the following comments and suggestions:
1) The paper is well written and clear. English, however, should be polished since there are several typos.
2) There is a typo in line 82, section 2 heading: kinematics should be Kinematics.
3) Section 2.2 (line 128) should Inverse Kinematic Analysis, not Forward Kinematics Analysis.
4) In line 130, it is said that the method for solving the inverse kinematics is the geometric method. There are several geometric methods so the authors should specify which one they are using or explain more in detail how they are obtaining such solutions. In addition, the method for solving the inverse kinematics based on inverting and multiplying the transformation matrices is Paul's method (and is not considered as a geometric method). Is that the one the authors are using?
5) In line 146, although it is correct to consider end-effector poses as configurations, to avoid confusion, they are termed as poses, while the term configuration is reserved for vectors of joint variables (thus defining the configuration space).
6) The paragraph of lines 155-160 should be explained in much more detail since it is not straightforward to see the relationship between the optimization goals and the three aspects the authors want to optimize.
7) In equation (9), the maximum velocity, acceleration and jerk are set as the default maximum ones of the robot. Since the robot is a collaborative one, would not be better if they are set in a particular value attending to the scenario and task?
8) In line 232, equation (18) is not aligned.
9) In table 3, since the trajectory defined is arbitrary or random, why the value of q4 is always zero? If a given trajectory has been designed to validate the method proposed, the authors should include it in the paper.
Reviewer 2 Report
Background - the references should be revised - the list seem to favour Chinese authors while the problem of trajectory planning has been discussed in many publications around the world.
Some items in the list of references are incorrectly formatted (like 5, 20, 25, 26, etc.)
Presentation of results: it would be more clear, if the authors provided a graph of the actual trajectory before and after optimization.
The values of weight coefficients (equation 21) are not given. Authors do not discuss the choice of the weights and do not provide alternative sets (leading most probably to alternative optimized solutions).
Conclusions simply outline the paper, there is no information on the authors' original contribution.
The conclusions seem to be based on just one example.
Abstract is identical with a part of the introduction (lines 67..77).
It is not proven how the proposed method "improves the operating efficiency of the collaborative robot" and "ensures that the trajectory is smooth and controllable" as claimed in the abstract.
Trajectory planning involves optimisation of speed, acceleration and jerk but there are only kinematic equations presented in the chapter 2. There is only one reference to robot's kinematics in chapter 3 (equation 5). How is the robot dynamics modelled in the experiment?
line 128 - Chapter 2.2 should be 2.3, the chapter title should also be corrected
Reviewer 3 Report
The paper refers to an interesting method for planning smooth trajectory for robots. The planning relies on seven-order B-spline under constraints to optimize jerk, acceleration and time. The manuscript is very well written and organized. Concepts are made clear before explaining the details. My comments on the article are as follows:
1] Line 82: please, use a capital letter to start the section title.
2] Line 89: include the name of the robot in the caption.
3] Line 100: I guess that authors used the robotics toolbox of Peter Corke, which must be cited.
4] Line 140: how do you consider the z Cartesian coordinates in the inverse kinematics? Which method do you apply to solve the problem?
Perhaps, you can have a look at this thesis:
'Robotic machining: Development and validation of a numerical model of robotic milling in order to optimise the cutting parameters' where inverse kinematics is explained in Appendix A.
5] Line 142: why theta_3 and theta_5 are giving multiple solutions? Isn't it theta_4 and theta_6 when both are axes are parallel?
6] Can this optimization be made online, or do we need to optimize the whole path on computer in advance ?
7] Line 341: How is that possible that joint 4 does not move for the chosen trajectory?
8] What are the future research plan to be included in the conclusion?
Are you gonna investigate the dynamics of the robot? Is the accuracy a concern with such a robot?
9] Figure 11: perhaps a link to a video would be great to actually see the robot moving on the optmized trajectory.
Round 2
Reviewer 1 Report
The authors have modified the manuscript accordingly with the suggestions of this reviewer, so, I recommend the editor to accept the paper for publication in its current form. I would like to thank the authors for the efforts in improving the paper and the answers provided to this reviewer.
Author Response
Dear Reviewer,
Thank you very much for reviewing this manuscript!
We really appreciate your valuable suggestions and comments.
Reviewer 2 Report
The paper has been significantly improved.
The references 16 and 17 should be corrected:
16 - the names of the authors are not correct : should be"Siciliano, B., Sciavicco, L., Villani, L., Oriolo, G.", not "L, S., S. B and V. L"
17 - should be "Corke, P.I." , not "P.I. Corke"
Author Response
Dear Reviewer,
Thank you very much for reviewing this manuscript!
We really appreciate your valuable suggestions and comments.
We have corrected the two references in the re-submitted paper.